# Antimicrobial Activity of Aztreonam in Combination with Old and New β-Lactamase Inhibitors against MBL and ESBL Co-Producing Gram-Negative Clinical Isolates: Possible Options for the Treatment of Complicated Infections

**DOI:** 10.3390/antibiotics10111341

**Published:** 2021-11-03

**Authors:** Gianluca Morroni, Raffaela Bressan, Simona Fioriti, Gloria D’Achille, Marina Mingoia, Oscar Cirioni, Stefano Di Bella, Aurora Piazza, Francesco Comandatore, Carola Mauri, Roberta Migliavacca, Francesco Luzzaro, Luigi Principe, Cristina Lagatolla

**Affiliations:** 1Department of Biomedical Sciences and Public Health, Polytechnic University of Marche, 60126 Ancona, Italy; g.morroni@staff.univpm.it (G.M.); s.fioriti@staff.univpm.it (S.F.); gloria.dachille@gmail.com (G.D.); m.mingoia@staff.univpm.it (M.M.); o.cirioni@staff.univpm.it (O.C.); 2Department of Life Sciences, University of Trieste, 34127 Trieste, Italy; rbressan@units.it (R.B.); clagatolla@units.it (C.L.); 3Clinical Department of Medical, Surgical and Health Sciences, University of Trieste, 34129 Trieste, Italy; stefano932@gmail.com; 4Unit of Microbiology and Clinical Microbiology, Department of Clinical-Surgical, Diagnostic and Pediatric Sciences, University of Pavia, 27100 Pavia, Italy; aurora.piazza@unipv.it (A.P.); roberta.migliavacca@unipv.it (R.M.); 5Department of Romeo and Enrica Invernizzi Pediatric Research Center, Department of Biomedical and Clinical Sciences L. Sacco, University of Milan, 20157 Milan, Italy; francesco.comandatore@unimi.it; 6Clinical Microbiology and Virology Unit, A. Manzoni Hospital, 23900 Lecco, Italy; c.mauri@asst-lecco.it (C.M.); f.luzzaro@asst-lecco.it (F.L.); 7Clinical Pathology and Microbiology Unit, S. Giovanni di Dio Hospital, 88900 Crotone, Italy

**Keywords:** aztreonam, β-lactamases inhibitors, synergism, MBLs, ESBL, complicated infection, difficult-to-treat pathogen

## Abstract

Metallo-β-lactamases (MBLs) are among the most challenging bacterial enzymes to overcome. Aztreonam (ATM) is the only β-lactam not hydrolyzed by MBLs but is often inactivated by co-produced extended-spectrum β-lactamases (ESBL). We assessed the activity of the combination of ATM with old and new β-lactamases inhibitors (BLIs) against MBL and ESBL co-producing Gram-negative clinical isolates. Six *Enterobacterales* and three non-fermenting bacilli co-producing MBL and ESBL determinants were selected as difficult-to-treat pathogens. ESBLs and MBLs genes were characterized by PCR and sequencing. The activity of ATM in combination with seven different BLIs (clavulanate, sulbactam, tazobactam, vaborbactam, avibactam, relebactam, zidebactam) was assessed by microdilution assay and time–kill curve. ATM plus avibactam was the most effective combination, able to restore ATM susceptibility in four out of nine tested isolates, reaching in some cases a 128-fold reduction of the MIC of ATM. In addition, relebactam and zidebactam showed to be effective, but with lesser reduction of the MIC of ATM. *E. meningoseptica* and *C. indologenes* were not inhibited by any ATM–BLI combination. ATM–BLI combinations demonstrated to be promising against MBL and ESBL co-producers, hence providing multiple options for treatment of related infections. However, no effective combination was found for some non-fermentative bacilli, suggesting the presence of additional resistance mechanisms that complicate the choice of an active therapy.

## 1. Introduction

Infections sustained by metallo-β-lactamases (MBLs)-producing bacteria pose a global challenge due to the paucity of effective antibiotic options. Despite that MBLs account only for 10% of the total β-lactamases [1], these enzymes are capable of hydrolyzing all current available β-lactams, with the exception of monobactams, such as aztreonam (ATM) [2]. Differently from serine-β-lactamases, MBLs are Zn-dependent metalloproteinases which are non-susceptible to inactivation by currently available β-lactams inhibitors (BLIs), including vaborbactam (VAB), a cyclic boronate, and avibactam (AVI), a diazabicyclooctanone (DBO) derivative [3]. The clinical impact of MBL-producing bacteria is rising, as evidenced by the recent outbreak of NDM-carrying *Enterobacterales* in Italy [4]. Of note, some non-fermentative Gram-negative bacilli naturally producing MBLs (*Chryseobacterium indologenes*, *Elizabethkingia meningoseptica*, and *Stenotrophomonas maltophilia*) are also intrinsically resistant or less susceptible to several antibiotics, including colistin [5,6].

The only β-lactam effective, because it is not hydrolyzed, against MBL-producers is ATM, a monobactam approved in 1986 but currently infrequently used due to the spread of extended-spectrum β-lactamases (ESBLs) [7]. ATM is effective against MBL-producer infections as long as these isolates are ESBL-negative. Many BLIs, namely clavulanic acid, sulbactam or tazobactam, are commonly used in clinical practice in association with β-lactams. However, none of them are useful agents against MBL-producers [8]. Due to the common association of MBL with other β-lactamases (especially ESBL and AmpC), a novel combination including ATM and AVI has been proposed but is not yet commercially available. AVI is a non-BLI that blocks β-lactamases through a reversible bond and is currently approved in association with ceftazidime for the treatment of complicated urinary tract infections (cUTI), intra-abdominal infections (IAI), pneumonia, and other infections sustained by Gram-negative bacteria [8]. Ceftazidime–AVI is ineffective against MBL-producers but the broad spectrum of activity of AVI against ESBLs suggested a possible application of its use if combined with ATM. Indeed, recent studies demonstrated a good efficacy of ATM–AVI combinations against *Enterobacterales*, including MBL-producers [9,10]. This association has been also used in the clinical settings with most of the cases showing a favorable outcome [7].

In addition to the ATM–AVI combination, novel BLIs in combination with β-lactams are currently being developed for the treatment of multidrug-resistant (MDR) Gram-negative infections. VAB, originally developed to inhibit KPC carbapenemases but also effective against several ESBLs, significantly lowered meropenem (MEM) MICs in both KPC-producing and non-KPC-producing carbapenem-resistant *Enterobacterales*, and the MEM–VAB combination was approved by the FDA in 2017 for the treatment of cUTI, including pyelonephritis, in adults [11]. Similarly, relebactam (REL) in combination with imipenem (IPM) is reported to be effective against ESBLs, KPC and AmpC producers and the combination IPM–REL was also approved in 2019 for the treatment of cUTI and IAI in adults [12]. Zidebactam (ZID) is a recently developed DBO derivative with a dual activity: in addition to the inhibition of β-lactamases (and differently from the other BLIs) it binds the Gram-negative PBP2 and retains a proper antibacterial activity [13]. ZID is currently being studied in association with cefepime with several data reporting the efficacy of this association against *Enterobacterales* and *Pseudomonas* [13,14,15], but data on its combination with other β-lactams are scant.

The aim of the present work was to assess the activity of ATM in association with both old (clavulanate, sulbactam and tazobactam) and new (AVI, REL, VAB, ZID) BLIs, to understand the antimicrobial activity of these combinations against different MBL- and ESBL-producers (including *Enterobacterales* and non-fermentative bacilli).

## 2. Results

### 2.1. Characterization of bla Genes

β-lactamase genes harbored by the study isolates, identified by polymerase chain reaction (PCR) or whole genome sequencing (WGS), are shown in Table 1. In particular, regarding acquired MBLs, two isolates harbored *bla*_NDM-1_ gene (*K. pneumoniae* KL 12 SG and *K. pneumoniae* LC954/14), one for *bla*_NDM-5_ (*E. coli* 482483), and three *bla*_VIM-1_ (*E. coli* CP-Ec3 and Cp-Ec4, *C. amalonaticus* N18). Of them, three isolates also harbored *bla*_TEM-1_ and *bla*_CTX-M-15_ (*E. coli* CP-Ec4, *E. coli* 482483, *K. pneumoniae* KL 12 SG), two *bla*_SHV-12_ (*E. coli* CP-Ec4, *C. amalonaticus* N18), and one *bla*_KPC-2_ (*E. coli* Cp-Ec3), while *K. pneumoniae* LC954/14 co-expressed *bla*_CTX-M__-__15_ and an allelic form of *bla*_SHV_, *bla*_SHV-182_, whose susceptibility to BLIs has never been investigated.

Among isolates with chromosomally encoded MBLs, *S. maltophilia* harbored L1 and L2 β-lactamases, *E. meningoseptica bla*_GOB-13_, *bla*_B-9_ and *bla*_CME-1_, and *C. indologenes bla*_CIA_.

### 2.2. Susceptibility Testing

#### 2.2.1. Checkerboard Assays

The activity of ATM alone and in combination with BLIs was first investigated using the broth microdilution. Results are presented in Table 1. The BLIs were used at concentrations recommended by the European Committee on Antimicrobial Susceptibility Testing (EUCAST), with the exception of ZID, for which this concentration has not yet been determined. ZID was initially tested as a stand-alone agent due to its dual mode of action, which includes both the inhibition of β-lactamases and a direct antibacterial activity. *C. indologenes*, *E. meningoseptica* and *S. maltophilia* were not inhibited by ZID 32 µg/mL, the highest dosage used in this study. In contrast, MIC values for the six *Enterobacterales* ranged from 1 to 8 µg/mL, so for testing in combination with aztreonam, zidebactam was used both at 0.5 µg/mL, which was a sub-inhibitory dose for all strains tested, and at a 1:1 ratio.

The addition of BLIs provided very different results between *Enterobacterales* and the non-fermentative bacilli. The MICs of ATM (MIC_ATM_) of both *C. indologenes* and *E. meningoseptica* were unaffected by the addition of any of the tested BLIs, leaving open the question of whether this was due to the intrinsic resistance of these bacteria to ATM, possibly due to the low affinity of the drug for the target protein PBP3, or to the inability of all BLIs to inhibit their chromosomally encoded serine-β-lactamases. In contrast, the β-lactamase L2 of *S. maltophilia* seemed overall the most susceptible to BLIs, with sulbactam and tazobactam lowering MIC_ATM_ to 4–8 µg/mL, VAB to 2 µg/mL, and AVI and REL, the most effective BLIs, lowered MIC_ATM_ to ≤1 µg/mL, which is consistent with restoration of ATM susceptibility.

A similar trend in the activity of the different BLIs was found in *Enterobacterales*, although some observed differences seemed to be related to the bacterial species rather than to the encoded α-lactamases. In fact, DBO derivatives were by far the most active agents and, among them, AVI showed the highest activity. This was particularly true for *C. amalonaticus*, where MIC_ATM_ was reduced below the ATM breakpoint after the addition of AVI or ZID, whereas REL was only slightly less effective; moreover, a partial increase of ATM susceptibility was detected in the two *K. pneumoniae* isolates also after the addition of VAB. Interestingly, the comparison between the two *K. pneumoniae* isolates underlined their different behavior after the addition of ZID, which caused a reduction of at least 128-fold in the MIC_ATM_ of *K. pneumoniae* KL 12 SG but did not affect that of *K. pneumoniae* LC954/14. Although this was not easily explained, it is worth noting that *K. pneumoniae* LC954/14 encodes the *bla*_SHV-182_ allelic form, whose susceptibility to various BLIs, least of all to ZID, has never been investigated.

Finally, the three *E. coli* strains exhibited a lower than expected reduction of their MIC_ATM_ after addition of all BLIs, suggesting the presence of additional resistance mechanisms. Checkerboard assays testing the combination aztreonam/zidebactam at a 1:1 ratio confirmed previous results, showing synergistic effect against *C. amalonaticus* N18, *K. pneumoniae* KL 12 SG and *S. maltophilia* but not against the three *E. coli*, *K. pneumoniae* LC954/14, *C. indologenes* and *E.* meningoseptica (Table 1).

#### 2.2.2. Time–Kill Assay on *Enterobacterales*

The activity of the DBO derivatives, which were the most effective in the microdilution assay, was further investigated by time–kill assay. Previous results were mostly confirmed, although in some cases it was necessary to increase the ATM concentration one- or two-fold to achieve comparable antimicrobial activity.

The combination of ATM with AVI proved to be the most effective one, determining a CFU reduction higher than 7 log_10_ (compared to the agents used alone) on both *C. amalonaticus* and *Klebsiella* strains, at concentrations of ATM below the breakpoint (Figure 1). The combination showed bacteriostatic activity against *K. pneumoniae* LC954/14 (2.7 log_10_ reduction from baseline after 24 h of incubation) and was bactericidal against the other two strains. In the case of the three *E. coli* strains, the efficacy of the combinations was partially confirmed, limited to the first 8 h of incubation (2.5–5.5 log_10_ reduction compared to the agents used alone), and thereafter a regrowth was detected. In this case, the concentration of ATM used in the combination was already above the breakpoint, so we decided not to repeat the test with higher doses since it would not be clinically relevant.

The efficacy of the combination ATM/REL was confirmed against *C. amalonaticus* and *K. pneumoniae.* REL was a less effective inhibitor than AVI, as expected, as indicated by the higher concentrations of ATM required to achieve the antimicrobial effect, which was equal to or higher than the ATM breakpoint. However, the combination was strongly bactericidal against *C. amalonaticus* (which was also most sensitive to AVI) while it was bacteriostatic for the two *Klebsiella* strains (Figure 2).

A peculiar trend was observed with ZID compared to the other DBO derivatives, most likely due to the dual mode of action of this drug. When used alone at concentrations below the MIC it caused an initial decrease in the number of viable cells followed by a regrowth, which reached levels comparable to the control in *K. pneumoniae* KL 12 SG and was slightly lower in *C. amalonaticus* (Figure 3). This was expected, as a consequence of the different susceptibility of the strains to this drug (Table 1) and could be ascribed to differences in the PBP2 target proteins. However, when ZID was used in combination with ATM, the two strains showed opposite behavior. The combination was synergistic against *K. pneumoniae* KL 12 SG after 24 h of incubation, with a strong bactericidal activity at ATM concentration below the breakpoint. In contrast, a much lower efficacy was shown against *C. amalonaticus.* Here, a synergistic effect was only observed at ATM concentration of 2 µg/mL, corresponding to a four-fold increase in the MIC determined by the checkerboard assay. Moreover, partial regrowth was detected after 24 h, even if the number of viable cells remained below the baseline. All these data suggest that ZID is a very poor inhibitor of *bla*_SHV-12_.

#### 2.2.3. Time–Kill Assay on *S. maltophilia*

In the case of *S. maltophilia*, all but one BLIs tested showed to be effective in lowering the MIC of ATM in the microdilution assay (Table 1). Those that restored aztreonam susceptibility were tested by time–kill assay as well. The efficacy was essentially confirmed for VAB, AVI and REL, although all the combinations required a two-four-fold increase of ATM concentration (Figure 4). AVI proved to be the most effective BLI (5 log_10_ reduction compared to the agents used alone), and REL, the other DBO derivative, was only slightly less effective. VAB, the DBO derivative, was found to be less active: it was necessary to increase ATM concentration to 8 µg/mL to observe a 2.5 log_10_ reduction compared to the agents used. All combinations had a bacteriostatic effect.

Finally, the activity of ZID was not confirmed in the time–kill assay, even at ATM concentration eight times higher than that detected in the checkerboard assay (data not shown). This result is not easily explained. It can be hypothesized that compared to the static incubation used in the checkerboard assay, the orbital shaking used in the time–kill provides a more oxygenated environment, which is more suitable for the growth of *Stenotrophomonas.* Therefore, the possibility that the previously observed synergistic effect between ATM and ZID is at least partially due to the difficult growth of *Stenotrophomonas* under low oxygen culture conditions must be considered.

#### 2.2.4. Time-Kill Assays with Aztreonam/Zidebactam 1:1 Ratio

Time-kill assays were also performed with aztreonam/zidebactam at 1:1 ratio against *C. amalonaticus* and *S. maltophilia*, while at 1:1 ratio was already synergistic in *K. pneumoniae* KL 12 SG with 0.5 mg/L concentration (Figure 3). Results were summarized in Appendix A. In *C. amalonaticus* synergies were also confirmed with aztreonam/zidebactam at 1:1 ratio of 1 mg/L. In *S. maltophilia*, although we did not detect synergies with 0.5 mg/L of zidebactam (regardless of aztreonam concentration), the two antibiotics showed synergistic effect with a 4 mg/L concentration, although the regrowth observed at 24 h suggests that this combination needs to be further analyzed in a higher number of *S. maltophilia* isolates. 

## 3. Discussion

Currently available β-lactams/BLIs combinations are frequently used to treat MDR infections but are not active against MBL-producing Gram-negatives. The growing incidence of infections sustained by MBL-producers highlights the urgency for new MBL-inhibitors [21]. Pending the development of new molecules, combinations of ATM with different BLIs are being investigated for the treatment of infections sustained by *Enterobacterales* expressing both serine-ß-lactamases and MBLs [22,23]. The combination of ATM with AVI was one of the most promising and has also been used in clinical settings with relatively good results [7]. Our data showed that, against MBL-producers, DBO derivatives were the most active BLIs when combined with ATM. Among them, AVI confirmed to be the most effective, restoring ATM susceptibility in four out of nine isolates, in some cases reaching a 128-fold reduction of ATM MIC. Thus, considering the breakpoint value for ATM, the combination ATM–AVI (1:4) could be considered for the treatment of infections caused by MBL- and ESBL-producing *Enterobacterales*. REL and ZID also showed inhibitory activity against the same strains (with the exception of ZID against *K. pneumoniae* LC954/14), but to a lesser extent, as they caused a smaller decrease in ATM MIC. An unexpected result was obtained with the three *E. coli* strains, which showed overall resistance to all ATM/BLIs combinations tested, although they encoded serine-ß-lactamases that would have been susceptible. Indeed, in the case of REL, which is known to block the *bla*_KPC-2_ activity [24], we expected the combination to be effective at least for *E. coli* CP-Ec3, but no inhibition was observed, suggesting that these strains carry some, as yet unidentified, additional resistance determinants. Either way, AVI confirmed to be the most potent inhibitor, lowering the MIC of ATM even in these strains, although the susceptibility was not restored, and it proved to be active in time–kill assays for only a few hours after exposure.

REL, reported to be effective against bacteria carrying KPC and ESBLs enzymes [12], is currently used in association with IPM/cilastatin for the treatment of Gram-negative cUTI and IAI [11]. Data on its association with ATM were limited to *Klebsiella* and *Stenotrophomonas* and reported synergies against ATM-resistant strains of 97.5% and 71%, respectively [25,26]. In our study synergy was confirmed not only in *K. pneumoniae* and *S. maltophilia*, but also in the *C. amalonaticus* isolate, where the time–kill curves showed marked bactericidal activity of ATM when associated either with AVI or with REL. In this context, however, it must be noted that this particular susceptibility might be related to the different genetic background with respect to β-lactamase in addition to the differences between species: *C. amalonaticus* carried only *bla*_SHV_ in addition to *bla*_VIM_, while the other isolates expressed a more complex variety of β-lactamases.

The newest developed BLI, ZID, is currently being studied in association with cefepime and showed good activity even against isolates carrying MBLs [15]. In the present study, it gave variable results, both when used alone and in combination with ATM, but this should probably be ascribed to its dual mode of action. Its efficacy when used as a standalone agent results from its affinity for PBP2 encoded by the different strains [13]. In this case, MICs ranged from >32 µg/mL for the non-fermentative bacilli to 1 µg/mL for the *E. coli* strains, with *C. amalonaticus* being slightly more resistant than *E. coli*. The two *K. pneumoniae* showed different behavior among themselves, but variability among *Klebsiella spp.* has already been reported and in some cases attributed to impermeability to the drug [15]. On the other hand, synergism with ATM is a consequence of its serine-ß-lactamases inhibition. In our experiments, the combination ATM–ZID proved synergistic in two strains, *C. amalonaticus* N18, which produced *bla*_SHV-12_, and *K. pneumoniae* KL 12 SG, which produced *bla*_TEM-1_ and *bla*_CTX-M-15_, but not in *K. pneumoniae* LC954/14, which in addition to *bla*_CTX-M-15_, also produced *bla*_SHV-182_, an allelic variant whose susceptibility to BLIs has never been investigated. This result, in conjunction with the susceptibility of the bacterium to ZID alone, which rules out the impermeability to the drug, strongly suggests that the *bla*_SHV-182_ variant is unlikely to be inhibited by ZID.

Interestingly, results of the ATM–ZID combination against *S. maltophilia* were extremely different when comparing checkerboard and time–kill assays. Although we did not find similar results for the other strains and although usually checkerboard and time–kill assays showed comparable results, it has already been reported that the different growth conditions may influence the final result [27].

Finally, *E. meningoseptica* and *C. indologenes* did not respond to any ATM–BLI combinations. The importance of these two species in nosocomial infections, especially pneumonia, is increasing, also considering their intrinsic resistance to antibiotics including β-lactams [28,29]. The presence of different β-lactamases could explain the non-susceptibility to any of the tested combinations, although the effective role of any enzymes in β-lactam resistance should be further investigated, as well as resistance determinants responsible for their low susceptibility to zidebactam that, at the moment, may only be hypothesized (e.g., PBP2 with low affinity for the drug, efflux pumps, poor permeability).

## 4. Materials and Methods

### 4.1. Strains, Culture Media and Chemicals

Nine Gram-negative clinical isolates (six *Enterobacterales* and three non-fermenting bacilli), collected from different Italian hospitals and previously described as MBL-producers (*bla*_VIM_, *bla*_NDM_ or chromosomally encoded MBLs) were selected on the basis of their serine-ß-lactamases gene contents, in order to study the antimicrobial activity of ATM in combination with old and new BLIs. In four of these isolates (two *Escherichia coli*, *Citrobacter amalonaticus* and *Klebsiella pneumoniae* LC954/14) these genes were already described [16,17,19]; the remaining five strains (*E. coli* 482483, *K. pneumoniae* KL 12 SG, *Chryseobacterium indologenes* LC650/17, *Elizabethkingia meningoseptica* LC596/11 and *Stenotrophomonas maltophilia* LC669/17) [18,20] were investigated for this feature as described below. The genotypic characteristics of all strains are summarized in Table 1.

Bacterial strains were routinely cultured aerobically in LB medium (Oxoid, ThermoFisher Scientific Inc., Basingstoke, UK) at 37 °C and stored at −80 °C. Susceptibility assays were carried out in Cation-Adjusted Müller Hinton Broth (CAMHB). Drugs were purchased from MedChemExpress (Sollentuna, Sweden) in powder form and dissolved either in dimethyl sulfoxide (DMSO) or in deionized water at a concentration of 2.5 mg/mL. Stock solutions were stored at −20 °C and properly diluted in CAMHB for each experiment.

### 4.2. Identification, Antimicrobial Susceptibility Testing

The identification at species level and antibiotic susceptibility testing were carried out using the MicroScan AutoScan4 System (Beckman Coulter, Brea, CA, USA). Bacterial identifications were confirmed by MALDI-TOF mass spectrometry (Vitek MS, BioMérieux, Marcy l’Etoile, France), analyzing colonies from McConckey agar (BioMérieux) (for *Enterobacterales* and *S. maltophilia*) or Columbia agar (sheep blood 5%, BioMérieux) (for *E. meningoseptica* and *C. indologenes*), incubated overnight at 37 °C. MICs were interpreted according to the 2020 EUCAST breakpoints [30].

### 4.3. Antimicrobial Resistance Genes Investigations

The genomic DNA was extracted using NucleoSpin Tissue (Macherey-Nagel, Dueren, Germany) kit. Check-MDR CT103XL (Check-Points Health B.V., Wageningen, The Netherlands) microarray was used for antimicrobial resistance genes investigation as described elsewhere [31]. PCR and sequencing were performed to assess the specific allelic variant of the beta-lactamase genes. The primers’ sequences and PCR conditions used for the detection of the *bla*KPC, *bla*VIM, *bla*NDM, *bla*CTX-M, *bla*TEM and *bla*SHV genes were run as previously described [9]. PCR products were purified using the quantum PrepPCR Kleen Spin Columns kit (ThermoFisher Scientific) and subjected to double-strand sequencing using the automatic sequencer ABI PRISM 3100 genetic analyzer DNA Sequencer (Applied Biosystems, Foster City, CA, USA) and the BigDye Terminator v1.1 Cycle Sequencing kit (Applied Biosystems, Foster City, CA, USA). The sequences were analyzed according to the BLAST software program (https://blast.ncbi.nlm.nih.gov/Blast.cgi, accessed on 1 February 2021).

### 4.4. Whole-Genome Sequencing

Two *E. coli* (CP-Ec3 and 482483), *K. pneumoniae* (LC954/14), *C. indologenes* (LC650/17), and *E. meningoseptica* (LC596/11) were investigated by WGS. The genomic DNA was extracted with the QIAamp DNA minikit (Qiagen, Germantown, MD, USA) following the manufacturer’s instructions. Library preparation was performed with Nextera XT library preparation kit (Illumina Inc., San Diego, CA, USA), and the sequencing using an Illumina Miseq platform (2 × 250 paired end run). Read quality was assessed using FastQC software (https://www.bioinformatics.babraham.ac.uk/projects/fastqc/, accessed on 1 April 2021), and reads were trimmed using Trimmomatic software [32] and assembled with SPAdes [33]. Resistance genes were investigated using the ResFinder online tool [34] and SRST2 software [35] with the ARG-ANNOT dataset [36].

### 4.5. MIC Evaluation and Checkerboard Assays

MIC of ATM, alone and in association with BLIs, was evaluated for each strain by broth microdilution, according to EUCAST guidelines [37]. Briefly, 5 × 10^5^ CFU/mL were inoculated in 100 µL of cation-adjusted Muller–Hinton broth containing 2-fold serial dilutions of the antimicrobial agent and incubated at 37 °C for 24 h. The MIC was recorded as the lowest concentration of the antimicrobial agent that inhibited visible growth. ATM was used as the maximum dose at 32 µg/mL (eight times the breakpoint), as higher doses were not considered clinically significant. BLIs were used at fixed concentrations recommended by EUCAST: clavulanate 2 µg/mL; tazobactam, sulbactam, AVI and REL 4 µg/mL; VAB 8 µg/mL. An association was considered synergistic if at least a four-fold decrease in the MIC of ATM was observed. Zidebactam (ZID), for which a recommended fixed concentration was not available, was tested in the range 0.125–32 µg/mL, alone and in association with ATM, by checkerboard assay in 96-well microtiter plates on an initial inoculum of 5 × 10^5^ CFU/mL. Besides, as the Clinical Laboratory and Standards Institute (CLSI) in the United States of America recommends that testing of zidebactam in combination with cefepime is carried out at a 1:1 ratio, additional testing of the combination aztreonam/zidebactam at a 1:1 ratio was carried out. The combination was considered synergistic when the fractional inhibitory concentration index (FICI) was ≤0.5. No interaction occurred when FICI was >0.5 and ≤4, and when FICI was >4 the effect was considered antagonistic [38].

### 4.6. Time–Kill Assay

The most active combinations of ATM–BLI were further evaluated for each strain. The assay was carried out in 2 mL broth samples, inoculated in a 24-well plate on an initial inoculum of 5 × 10^5^ CFU/mL and incubated at 37 °C with gentle agitation. Positive controls in CAMHB without drug and in non-inhibitory concentrations of ATM and BLI alone were always included. Viable cells were evaluated by plating serial dilutions at 0, 4, 8 and 24 h. Time–kill curves were generated by plotting the mean colony counts of three independent experiments (log_10_ CFU/mL) ± standard deviation. Bactericidal activity was defined as a 3 log_10_ CFU/mL reduction from baseline at 24 h. Synergy between two agents was defined as a 2 log_10_ CFU/mL reduction compared with the most active agent alone [38].

## 5. Conclusions

The activity of different BLIs in combination with ATM against MBL- and ESBL-producers was evaluated. AVI confirmed to be the most active BLI, also in comparison with the newly developed ZID, which has been studied mainly in combination with cefepime but for which few data are available in combination with ATM. The study confirmed the overall antimicrobial activity of the ATM–BLI combinations, but also revealed the failure of some combinations against certain isolates.

Certainly, this study has some important limitations, particularly with respect to the small number of isolates selected and the lack of a detailed description of the resistance mechanisms involved. However, it provides results that could be a first step in the evaluation of ATM–BLIs combinations as an alternative to ATM–AVI for the treatment of difficult-to-treat infections caused by MBL- and ESBL-producers. It highlights the importance of a good knowledge of the activity of each ATM–BLI combination against the different ß-lactamases in order to select the better combination and avoid therapeutic failures as much as possible.

In addition, the study brought to light some resistance determinants (e.g., the ß-lactamase SHV-182 and the PBP-2) that are still poorly known but probably have an important impact on the activity of BLIs, thus requiring further investigation in the near future.

## Figures and Tables

**Figure 1 antibiotics-10-01341-f001:**
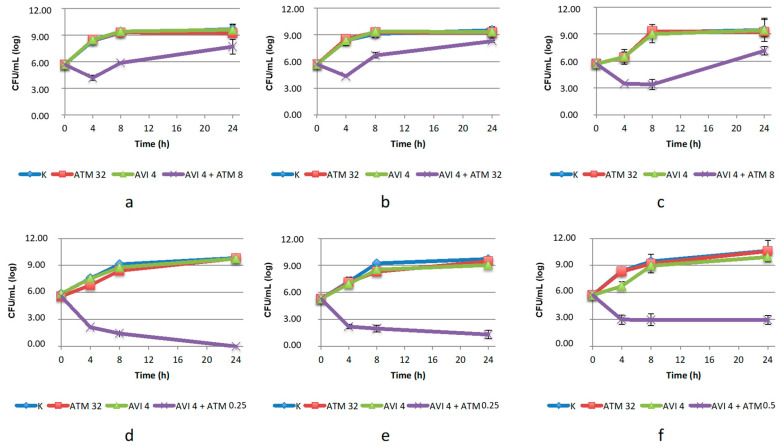
Twenty-four-hour time–kill curves of aztreonam (ATM) and avibactam (AVI), alone and in combination, on: (**a**) *E. coli* CP-Ec3; (**b**) *E. coli* CP-Ec4; (**c**) *E. coli* 482483; (**d**) *C. amalonaticus* N18; (**e**) *K. pneumoniae* KL 12 SG; (**f**) *K. pneumonia* LC954/14. K: positive control (without the addition of antimicrobials). Mean values and standard deviation of three independent experiments are reported.

**Figure 2 antibiotics-10-01341-f002:**
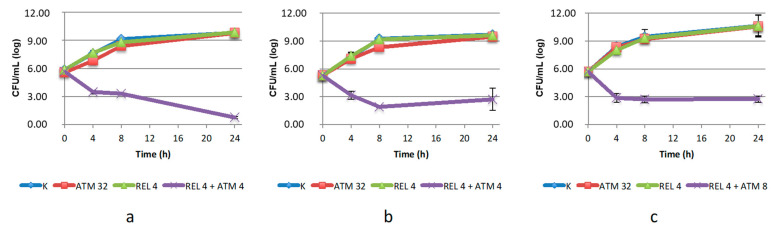
Twenty-four-hour time–kill curves of aztreonam (ATM) and relebactam (REL), alone and in combination, on: (**a**) *C. amalonaticus* N18; (**b**) *K. pneumoniae* KL 12 SG; (**c**) *K. pneumoniae* LC954/14. K: positive control (without the addition of antimicrobials). Mean values and standard deviation of three independent experiments are reported.

**Figure 3 antibiotics-10-01341-f003:**
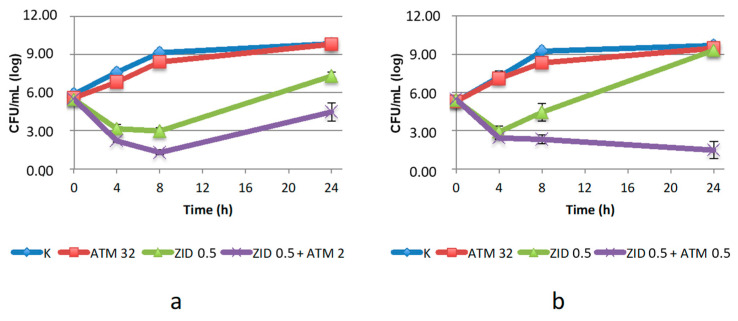
Twenty-four-hour time–kill curves of aztreonam (ATM) and zidebactam (ZID), alone and in combination, on: (**a**) *C. amalonaticus* N18; (**b**) *K. pneumoniae* KL 12 SG. K: positive control (without the addition of antimicrobials). Mean values and standard deviation of three independent experiments are reported.

**Figure 4 antibiotics-10-01341-f004:**
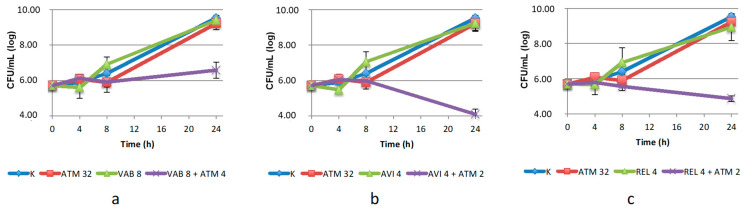
Twenty-four-hour time–kill curves of aztreonam (ATM) and: (**a**) vaborbactam (VAB); (**b**) avibactam (AVI); (**c**) relebactam (REL), alone and in combination, against the *S. maltophilia* isolate. K: positive control (without the addition of antimicrobials). Mean values and standard deviation of three independent experiments are reported.

**Table 1 antibiotics-10-01341-t001:** MIC (µg/mL) of aztreonam, alone and in association with BLIs, of tested strains.

Strain	Metallo-β-Lactamase	Serine-β- Lactamase	MIC ATM	MIC ATM after Addition of							MIC ZID	MIC ATM/ZID 1:1 Ratio	REF
				CLA ^a^	TAZ ^b^	SUL ^b^	VAB ^c^	AVI ^b^	REL ^b^	ZID ^d^			
** *E. coli* CP-Ec3**	*bla* _VIM-1_	*bla* _KPC-2_	>32	>32	>32	>32	>32	8	>32	32	1	1	[16]
** *E. coli* CP-Ec4**	*bla* _VIM-1_	*bla* _TEM-1_ * *bla* _CTX-M-15_ * *bla* _SHV-12_	>32	16	>32	>32	>32	32	>32	>32	1	1	[16]
** *E. coli* 482483**	*bla* _NDM-5_	*bla* _TEM-1_ * *bla* _CTX-M-15_	>32	16	>32	>32	>32	8	>32	32	1	1	[17]
** *C. amalonaticus* N18**	*bla* _VIM-1_	* *bla* _SHV-12_	>32	8	>32	>32	>32	0.25	4	0.5	4	0.5	[18]
** *K. pneumoniae* ** **KL 12 SG**	*bla* _NDM-1_	*bla* _TEM-1_ * *bla* _CTX-M-15_	>32	>32	>32	>32	4	0.25	4	0.25	8	0.5	This study
** *K. pneumoniae* LC954/14**	*bla* _NDM-1_	* *bla* _CTX-M-15_ * *bla* _SHV-182_	>32	>32	>32	>32	8	0.25	4	>32	1	1	[19]
** *C. indologenes* LC650/17**	*bla* _IND-3_	* *bla* _CIA_	>32	>32	>32	>32	>32	>32	>32	>32	>32	>32	[20]
** *E. meningoseptica* LC596/11**	*bla* _B-9_ *bla* _GOB-13_	* *bla* _CME-1_	>32	>32	>32	>32	>32	>32	>32	>32	>32	>32	[20]
** *S. maltophilia* **	*bla* _L-1_	* *bla* _L-2_	>32	>32	4	8	2	1	0.5	0.5	>32	0.5	[20]

MIC, minimum inhibitory concentration; BLIs, β-lactamase inhibitors; ATM, aztreonam; CLA, clavulanate; TAZ, tazobactam; SUL, sulbactam; VAB, vaborbactam; AVI, avibactam; REL, relebactam; ZID, zidebactam. a: 2 µg/mL; b: 4 µg/mL; c: 8 µg/mL; d: 0.5 µg/mL. * ESBL.

## Data Availability

The data presented in this study are available on request from the corresponding author.

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
