# Peer review of "Antimicrobial Activity of Aztreonam in Combination with Old and New β-Lactamase Inhibitors against MBL and ESBL Co-Producing Gram-Negative Clinical Isolates: Possible Options for the Treatment of Complicated Infections"

_antibiotics, 2021, doi:10.3390/antibiotics10111341_

Round 1
Reviewer 1 Report
The study is quite impressive and scientifically elaborated. I advise some minor corrections:
- In the material and methods section, please add the description of all protocols (parameters and conditions of analysis to make the tests more visible for lecturers):
* MALDI-TOF identification
* MICs determination
* give more details how do you perform the antimicrobial resistance gene.
- Why do you use the 32 µg/m of ATM as maximum dose? Explain in the text.
- I suggest for authors to verify the parameters and conditions of the analysis and the machine information (name, manufacturers ...).
- Please, improve the quality of all figures.
- In table 1, what is the meaning of the small letter existing at the top of the names of antibiotics (CLAa, TAZb, SULb, ...).
- Ensure the uniformity in the units (e.g. mg/mL, µL) throughout the MS.
- Correct some English mistakes before publication.
- Check the references in accordance with the journal style.
Author Response
Reviewer 1
The study is quite impressive and scientifically elaborated. I advise some minor corrections:
- In the material and methods section, please add the description of all protocols (parameters and conditions of analysis to make the tests more visible for lecturers):
- MALDI-TOF identification
The MALDI-TOF identification process has been described more in detail, as required.
- MICs determination
A brief description of the microbroth dilution method was added.
- give more details how do you perform the antimicrobial resistance gene.
We condensed this section in order to not make heavy the text. Hence, detailed protocols for PCR were fully described by Refs #9 and #31, while for WGS, by Refs from #32 to #36.
- Why do you use the 32 µg/m of ATM as maximum dose? Explain in the text.
An explanation was added in the text
- I suggest for authors to verify the parameters and conditions of the analysis and the machine information (name, manufacturers ...).
Parameters and conditions have been verified and machine information have been added where appropriate.
- Please, improve the quality of all figures.
The quality of the figures included in the main text was improved.
- In table 1, what is the meaning of the small letter existing at the top of the names of antibiotics (CLAa, TAZb, SULb, ...).
Sorry, it was a mistake. They have been deleted.
- Ensure the uniformity in the units (e.g. mg/mL, µL) throughout the MS.
All units have been uniformed throughout the text.
- Correct some English mistakes before publication.
English has been revised throughout the entire MS.
- Check the references in accordance with the journal style.
The check of references has been done.
Reviewer 2 Report
Comments to the Author:
MBLs are a serious problem in the treatment of Gram-negative infections worldwide. Therefore, the development of new β-lactam/β-lactamase inhibitor combinations is necessary. Morroni et al. aimed to investigate the antimicrobial activity of different combinations aztreonam/β-lactamase inhibitors against MBL and SBL co-producing Gram-negative strains. The authors have selected the inhibitors clavulanic acid, tazobactam, sulbactam, vaborbactam, avibactam, relebactam and zidebactam (clinical phase III). The combinations aztreonam/β-lactamase inhibitors were tested against six Enterobacterales strains and three non-fermenting bacilli strains by broth microdilution and time-kill curves. Aztreonam is the only β-lactam that is not affected by MBLs, so this manuscript examines the ability of the different inhibitors to inhibit the SBLs co-produced by the strains. Although the subject of the manuscript is interesting, it fails to add new information to what is already known, AZT/AVI is a promising combination. I have the following comments for the author’s consideration,
Major comments:
- The number of strains is low. For some species, only one strain is shown, so it is difficult to conclude from the results obtained. This is an important limitation of the work.
- Line 123-125. The authors indicate that there are no EUCAST recommended values for the use of zidebactam in microdilution assays. This is true. However, most published studies tested zidebactam at a 1:1 ratio in combination with cefepime. Furthermore, the most recent version of the CLSI criteria (M100 2021) also recommends this ratio for susceptibility testing. Re-determination of MICs to AZT/ZID is necessary.
- The antimicrobial activity of AVI alone should be determined against all strains. Some antimicrobial activity of this inhibitor against E. coli strains has been reported (doi: 10.1128/AAC.01585-15).
- Lines 131-157. The resistance mechanisms involved are not studied in detail. This is another important limitation of the work. For example, the high activity of vaborbactam against K. pneumoniae or the different activity of zidebactam between strains in the same species should be further analysed. One option that would considerably improve the work could be the cloning of the most interesting SBLs (CTX-M-15, KPC-2, etc.) in a reference strain and the determination of MICs. Searching for alterations in PBPs or porins would also be a good option.
- I recognize the good and hard work on the determination of time-kill curves, but I don't think these trials have provided more information than the MICs in this study.
Minor comments:
Abstract:
- Line 31. "Enterobacterales" is not italicized, it is an order.
- Line 37. change "ATMMIC" to the full word.
Introduction:
- Line 81. Although zidebactam is not an inhibitor of MBLs, it shows activity against MBL-producing strains. On the other hand, the inhibitor called taniborbactam (in clinical phase III in combination with cefepime) shows activity against MBLs. I recommend deleting "but without activity against MBL producers".
Results
- Line 121. Change “microbroth dilution” by “broth microdilution”.
- Line 129-130. As I indicated previously, zidebactam should be re-evaluated. Susceptibility to a 1:1 AZT-ZID ratio must be performed.
- Line 143. Change “beta lactamases” by “β-lactamases”.
- Lines 168-173. Is there an explanation for what has been observed in E. coli?
- Line 197. This is an assumption, to know the activity of zidebactam against the different PBP2 of these strains other assays would have to be performed.
- Lines 205-206. Without kinetic experiments or the cloning of SHV-12 in a reference strain, it is difficult to make this claim.
- Lines 224-226. It would be necessary to propose an explanation for this phenomenon.
Discussion:
- Lines 252-253. Two E. coli were sequenced. Indicate whether any further mechanisms of β-lactam resistance have been identified in them.
- Lines 287-297. As I indicated previously, I think it is difficult to conclude from the results shown by a single strain. Including more strains of the same species would be highly recommended.
Materials and Methods:
- Lines 347-348. As the previous comment, it would be interesting to indicate the mechanisms of β-lactam resistance found in the sequencing of these strains. Perhaps sequencing of the other strains would also be of interest.
- Lines 168-169. The values of FICI should be shown, at least the most interesting ones.
Tables
- Table 1. Indicate concentrations of MIC values (mg/L?). Indicate what is CLAa, TAZb, etc. Indicate which enzymes are ESBLs.
Figures
- If possible I would recommend increasing the size of all figures and the quality. Some of them look a bit pixelated.
Author Response
Reviewer 2
Comments to the Author:
MBLs are a serious problem in the treatment of Gram-negative infections worldwide. Therefore, the development of new β-lactam/β-lactamase inhibitor combinations is necessary. Morroni et al. aimed to investigate the antimicrobial activity of different combinations aztreonam/β-lactamase inhibitors against MBL and SBL co-producing Gram-negative strains. The authors have selected the inhibitors clavulanic acid, tazobactam, sulbactam, vaborbactam, avibactam, relebactam and zidebactam (clinical phase III). The combinations aztreonam/β-lactamase inhibitors were tested against six Enterobacterales strains and three non-fermenting bacilli strains by broth microdilution and time-kill curves. Aztreonam is the only β-lactam that is not affected by MBLs, so this manuscript examines the ability of the different inhibitors to inhibit the SBLs co-produced by the strains. Although the subject of the manuscript is interesting, it fails to add new information to what is already known, AZT/AVI is a promising combination. I have the following comments for the author’s consideration,
Major comments:
- The number of strains is low. For some species, only one strain is shown, so it is difficult to conclude from the results obtained. This is an important limitation of the work.
We agree with Reviewer’s comment. However, the aim of the study was to obtain a first preliminary evaluation of the antimicrobial activity of the different ATM/BLI combinations against selected strains producing both MBLs and ESBLs and isolated from the real-life context. Each strain was selected taking in consideration different combinations of species and resistance profiles. To this regard, our study could be useful in light of further studies, including more isolates harboring other resistance determinants (e.g. IMP determinants). This caveat has been added in the Conclusions section.
- Line 123-125. The authors indicate that there are no EUCAST recommended values for the use of zidebactam in microdilution assays. This is true. However, most published studies tested zidebactam at a 1:1 ratio in combination with cefepime. Furthermore, the most recent version of the CLSI criteria (M100 2021) also recommends this ratio for susceptibility testing. Re-determination of MICs to AZT/ZID is necessary.
Yes, we are aware of these recommendations. However, the aim of the present study was to preliminaraly test whether there is a synergistic effect between aztreonam and zidebactam, a combination that, up to now, has been little studied. Testing different dosage combinations for this purpose will, in our opinion, provide a more complete picture of the interaction between the two antimicrobials.
- The antimicrobial activity of AVI alone should be determined against all strains. Some antimicrobial activity of this inhibitor against E. coli strains has been reported (doi: 10.1128/AAC.01585-15).
We thank the Reviewer for this comment and are aware that in some cases avibactam may have antimicrobial activity against clinical isolates as a sole agent. However, we can confirm with certainty that this is not the case for the strains investigated in this study as time-kill experiments show that the addition of avibactam 4 µg/mL to the culture medium does not affect the growth rate at all compared to the control.
- Lines 131-157. The resistance mechanisms involved are not studied in detail. This is another important limitation of the work. For example, the high activity of vaborbactam against K.pneumoniae or the different activity of zidebactam between strains in the same species should be further analysed. One option that would considerably improve the work could be the cloning of the most interesting SBLs (CTX-M-15, KPC-2, etc.) in a reference strain and the determination of MICs. Searching for alterations in PBPs or porins would also be a good option.
We agree with the Reviewer’s comment. As mentioned in point #1, our study would represent a first preliminary step indication of the antimicrobial activity of ATM/BLI combinations against selected MBLs and ESBLs-producing isolates, based on detailed phenotypic results. The evaluation of all resistance determinants in all strains by cloning, although very meaningful and informative, could constitute another stand-alone study, as already mentioned in point #1. Moreover, we would recruit (and contact) another working team for cloning experiments in the near future. As mentioned in point #1, this limitation has been added in the Conclusions section.
- I recognize the good and hard work on the determination of time-kill curves, but I don't think these trials have provided more information than the MICs in this study.
In this case we disagree with the Reviewer. Time-kill curves provide a dynamic picture of antimicrobial activity, and make it possible to understand immediately the type of activity, bacteriostatic or bactericidal, of the different combinations and, above all, to show that effective combinations are effective after a short time (often after only 4 hours), an aspect that is not negligible for the pharmacodynamics of the therapy.
Minor comments:
Abstract:
- Line 31. "Enterobacterales" is not italicized, it is an order.
OK, it has been corrected.
- Line 37. change "ATMMIC" to the full word.
Ok, it has been changed.
Introduction:
- Line 81. Although zidebactam is not an inhibitor of MBLs, it shows activity against MBL-producing strains. On the other hand, the inhibitor called taniborbactam (in clinical phase III in combination with cefepime) shows activity against MBLs. I recommend deleting "but without activity against MBL producers".
Ok, it has been deleted.
Results
- Line 121. Change “microbroth dilution” by “broth microdilution”.
OK, it has been changed.
- Line 129-130. As I indicated previously, zidebactam should be re-evaluated. Susceptibility to a 1:1 AZT-ZID ratio must be performed.
For the aim of the present study, testing different dosage combinations will, in our opinion, provide a more complete picture of the interaction between the two antimicrobials.
- Line 143. Change “beta lactamases” by “β-lactamases”.
OK, it has been changed.
- Lines 168-173. Is there an explanation for what has been observed in E. coli?
We did not have any data to give a possible explanation as we did not find other β-lactam resistance mechanisms that could explain these behaviors.
- Line 197. This is an assumption, to know the activity of zidebactam against the different PBP2 of these strains other assays would have to be performed.
We agree with the Reviewer’s comment. Nevertheless, to emphasize that this is only a hypothesis that still needs a confirmation, we add a comment in the new Conclusion section.
- Lines 205-206. Without kinetic experiments or the cloning of SHV-12 in a reference strain, it is difficult to make this claim.
Certainly it is difficult to claim. But for this very reason we confine ourselves to suggesting a possible explanation, we have not made an assertion. As already said above, a specific study of this allelic form was beyond the aim of this study. Nevertheless, to emphasize that this is only a hypothesis that still needs a confirmation, we add a comment in the new Conclusion section.
- Lines 224-226. It would be necessary to propose an explanation for this phenomenon.
Culture conditions differed slightly between the checkerboard assay (static incubation) and the time-kill assay (orbital shaking). Stenotrophomonas is a non-fermentative bacillus for which aerobic metabolism is much more important than for Enterobacterales. It is possible that in the checkerboard assay the low oxygen supply limits the growth of Stenotrophomonas, making it more sensitive than it actually is. A comment was added in the MS.
Discussion:
- Lines 252-253. Two E. coli were sequenced. Indicate whether any further mechanisms of β-lactam resistance have been identified in them.
We did not find any other β-lactam resistance mechanisms besides those reported in table 1.
- Lines 287-297. As I indicated previously, I think it is difficult to conclude from the results shown by a single strain. Including more strains of the same species would be highly recommended.
As mentioned in points #1 and #4, this represents a limitation of the study and it has been added in the discussion section.
Materials and Methods:
- Lines 347-348. As the previous comment, it would be interesting to indicate the mechanisms of β-lactam resistance found in the sequencing of these strains. Perhaps sequencing of the other strains would also be of interest.
Regarding the sequenced strains we did not find any other β-lactam resistance mechanisms. Sequencing of the other strains and evaluation of the contribute to resistance of all mechanisms will be the aim of the subsequent study.
- Lines 168-169. The values of FICI should be shown, at least the most interesting ones.
FICI values of the combination ATM-ZID were added in Table 1. Other BLIs were used at concentration suggested by EUCAST without evaluating the original MIC values, thus FIC indexes were not calculated.
Tables
- Table 1. Indicate concentrations of MIC values (mg/L?). Indicate what is CLAa, TAZb, etc. Indicate which enzymes are ESBLs.
MIC values and ESBL enzymes has been indicated. The small letters at the top of the names of antibiotics were a mistake and have been deleted.
Figures
- If possible I would recommend increasing the size of all figures and the quality. Some of them look a bit pixelated.
The quality of the figures included in the main text was improved.
Reviewer 3 Report
This manuscript describes on the activity of the combination of aztreonam and a β-lactamase inhibitor (BLI) against MBL / ESBL communist Gram-negative clinical isolates, but I have some comments to the author.
Comment 1. What do the small font a, b, c, d in Table 1's MIC ATM after addition of… mean?
Comment 2. What is the strain in (f) of Figure 1? I don't think there is an explanation in the Figure legend.
Comment 3. What is the blue line “K” in each figure?
Comment 4. In each figure, the concentration of aztreonam (ATM) used in the combination is different, which is confusing to read. For example, in Figure 4, in (a), VAB 8 µg/ml is ATM 4 µg/ml, while in (b) and (c), AVI 4 µg/ml and REL 4 µg/ml are ATM 2 µg/ml, respectively. is. I didn't quite understand the ATM concentration setting I used.
Comment 5. From this study, I got the impression that the combination of avibactam (AVI) and ATM is the most effective, but I would like you to recommend clinically which concentration combination of AVI and ATM should be used.
Author Response
Reviewer 3
This manuscript describes on the activity of the combination of aztreonam and a β-lactamase inhibitor (BLI) against MBL / ESBL communist Gram-negative clinical isolates, but I have some comments to the author.
Comment 1. What do the small font a, b, c, d in Table 1's MIC ATM after addition of… mean?
Comment 2. What is the strain in (f) of Figure 1? I don't think there is an explanation in the Figure legend.
We are sorry, both of them were a mistake and have been corrected. Thank you.
Comment 3. What is the blue line “K” in each figure?
It is the positive control (without the addition of antimicrobials). It has been added in the legend
Comment 4. In each figure, the concentration of aztreonam (ATM) used in the combination is different, which is confusing to read. For example, in Figure 4, in (a), VAB 8 µg/ml is ATM 4 µg/ml, while in (b) and (c), AVI 4 µg/ml and REL 4 µg/ml are ATM 2 µg/ml, respectively. is. I didn't quite understand the ATM concentration setting I used.
Concentrations of ATM were selected based on checkerboard results that identified the highest synergistic combinations. The different ATM concentrations required to achieve synergistic effects are the direct consequence of the different efficiencies of the BLIs tested: the higher the MIC of aztreonam, the lower the activity of the inhibitor, as commented in the Results section.
Comment 5. From this study, I got the impression that the combination of avibactam (AVI) and ATM is the most effective, but I would like you to recommend clinically which concentration combination of AVI and ATM should be used.
A comment was added in the discussion.
Round 2
Reviewer 2 Report
The authors have responded satisfactorily to all my comments. They have clarified the message they want to convey in this paper. I think the manuscript has been improved and may be a good starting point for the study of new combinations of aztreonam and new beta-lactamase inhibitors.
Author Response
The authors have responded satisfactorily to all my comments. They have clarified the message they want to convey in this paper. I think the manuscript has been improved and may be a good starting point for the study of new combinations of aztreonam and new beta-lactamase inhibitors.
Thank you for your precious comments.